# MLD: An Intelligent Memory Leak Detection Scheme Based on Defect Modes in Software

**DOI:** 10.3390/e24070947

**Published:** 2022-07-07

**Authors:** Ling Yuan, Siyuan Zhou, Peng Pan, Zhenjiang Wang

**Affiliations:** Department of Computer Science, Huazhong University of Science and Technology, Wuhan 430074, China; siyuanzhou@163.com (S.Z.); panpeng@hust.edu.cn (P.P.); M202073292@hust.edu.cn (Z.W.)

**Keywords:** software defect detection, memory leak detection algorithm, software detection, defect modes

## Abstract

With the expansion of the scale and complexity of multimedia software, the detection of software defects has become a research hotspot. Because of the large scale of the existing software code, the efficiency and accuracy of the existing software defect detection algorithms are relatively low. We propose an intelligent memory leak detection scheme MLD based on defect modes in software. Based on the analysis of existing memory leak defect modes, we summarize memory operation behaviors (allocation, release and transfer) and present a state machine model. We employ a fuzzy matching algorithm based on regular expression to determine the memory operation behaviors and then analyze the change in the state machine to assess the vulnerability in the source code. To improve the efficiency of detection and solve the problem of repeated detection at the function call point, we propose a function summary method for memory operation behaviors. The experimental results demonstrate that the method we proposed has high detection speed and accuracy. The algorithm we proposed can identify the defects of the software, reduce the risk of being attacked to ensure safe operation.

## 1. Introduction

With the widespread use of computer technology in software security and processing, the scale of software is growing rapidly. Therefore, defects in the software are inevitable, some of which are fatal and will endanger the safety of the data [1]. The detection algorithm can reduce the bug of software, which can reduce the risk of being maliciously attacked. The sustained occurrence of memory leaks will cause a sharp decrease in the available memory, which will affect the system capability and even cause server downtime. Therefore, memory leak detection algorithms are urgently needed to detect defects. Compared with dynamic detection, static memory leak detection has some advantages: it can detect defects in the early stage of program development and reduce the cost of maintenance in subsequent stages. It can detect defects in the software before it is started. Test cases do not need to be artificially designed, which solves the software testing limitation caused by incomplete test cases. However, this method also has the disadvantages of slow detection speed and severe false positives [2]. Therefore, reducing the False Positive Rate of memory leak detection to improve the detection efficiency is the focus of this paper.

The contributions of this paper are described as follows:(1)We summarize and analyze the common memory leak defect modes in C/C++ and design a fuzzy matching algorithm based on regular expression. We analyze the summarized defect modes by simple lexical analysis, and then we classify the lexical units into variables, keywords and numbers to form a series of defect strings in form of regular expressions to be matched. We employ auxiliary methods such as path sensitivity analysis, interval arithmetic and alias analysis to improve the accuracy of memory leak detection and reduce the False Positive and False Negative Rates.(2)We summarize memory operation behaviors and classify them into three types based on the memory leak defect modes: allocation, deallocation and transfer. We design the state machine to detect the possible memory operation behaviors in the program using the fuzzy matching algorithm based on regular expression. We detect the memory leak defect by controlling the state change of the state machine model-based memory operation behavior.(3)To improve the efficiency of detection and reduce the scan times of program functions, we propose a function summary method for memory operation behaviors to achieve a brief description of memory operation behaviors in functions. Static detection methods based on a state machine are divided into a general method and a special method. General memory leak detection is combined with a function summary and a Control Flow Graph (CFG) based on the State Machine to achieve memory leak detection. The special detection is static detection based on rules.

The paper is organized as follows: Section 2 presents related work. In Section 3, we discuss the system model and definitions. Section 4 discusses our proposed MLD scheme. In Section 5, we present experiments and performance analysis. In Section 6, we analyze the prospects for industrial application of the proposed method. Finally, Section 7 discusses the conclusion and future work.

## 2. Related Work

For memory leak detection, dynamic detection [3,4,5,6] involves executing test cases and identifying potential memory leak defects in the program to be detected, such as LeakPoint [3], valgrind [4], Purify [5] and Maebe [7]. Their disadvantages include an overly high False Negative Rate and excessive dependence on the inputted use cases, which can not be used in continuously running software. In the past few years, machine learning algorithms are employed in memory leak detection. The process is to first obtain datasets through experiments and then to compute feature vectors and train a model based on data, such as GenCount [8], LBRCPD+PrecogMF [9] and DEF-LEAK [10]. Their main disadvantage is the huge cost in size of parameters and time of training [11].

The static detection of a memory leak utilizes static analysis technology (symbolic execution, context sensitivity analysis, and function summary generation) to detect potential memory leak defects in a program without executing the program. Therefore, it is suitable for defect detection of large-scale software that requires continuous operation.

Fan Gang and Wu et al. [12] proposed a memory detection model for C/C++ based on a staged analysis of use-flow graph and path feasibility: SMOKE. Similarly, an automated approach proposed by Bhavana, Veena and Sahu [13], has two-staged detection of memory leaks based on time threshold analysis. Yu Lei and Ding et al. [14,15] designed a memory leak detection tool based on value-flow analysis: SABER. MEMLOCK [16] proposed by Wen Cheng and Wang Haijun et al, is an enhanced grey-box technique that is implemented according to both coverage and memory consumption information. Xu [17] proposed the Memory State Transition Graph (MSTG) and implemented the tool Melton. However, the above solutions are not sufficient in terms of detection accuracy, and are difficult to adapt to large-scale software defect detection tasks.

## 3. System Model and Definitions

We propose an intelligent memory leak detection scheme MLD based on defect modes in software, which belongs to static memory leak detection method. We propose matching algorithm to obtain potential defect statements in the source code. In addition, we combine relevant auxiliary techniques to control state changes in the defect modes state machine. We find potential defects by analyzing the current state information and get the defets reports, as shown in Figure 1. The core idea is detailed as follows:(1)Defect Modes Matching: Using a matching algorithm to obtain the defect points (memory operation behavior) that can be used to control the state change of the state machine from the program to be detected.(2)Defect Modes State Machine: Using the matched defect points to control the state change in the defect modes state machine to obtain potential defects in the program.(3)Auxiliary Techniques: This part employs related techniques with state machine to the detection of the defects, which primarily includes the function summary and CFG.

### 3.1. Memory Leak Defect Modes Analysis

The software defect modes [18] are summaries of certain types of defects or errors that often occur in a program, including the forms of the defects and the conditions of their generation. We analyze common memory leaks to better match software memory defects. Common memory leak defect modes include missing release memory leak, pointer memory leak, memory leak of mismatched request and release, and class member memory leak. For memory leak detection based on defect patterns, first, we need to preprocess the program through lexical analysis, and then generate a double-linked list of lexical units. Secondly, we create the symbol table of the entire file through interval division, and combine the attributes of the lexical unit node with its scope to establish the symbol table in the scope.

The lexical unit [19] is the smallest description of a single keyword in the source code, which record the attribute information of a single lexical unit. The structure of the doubly linked list is shown in Figure 2.

We use a symbol table [20,21] to record the program scope and related information within the scope, which is a supplement to the lexical unit attributes. The structure design of the symbol table is shown in Figure 3.

The programs are different due to the different structures of different programs. Matching the general defect patterns with the special programs is a problem that needs to be solved in the defect pattern matching process. We use a summary of C/C++ to statistically analyze the keyword categories in the program, including variables, numbers, and operators. A summary of the keywords in the defect modes is provided in Table 1.

### 3.2. Memory Operation Behaviors Analysis

The analysis of the memory operation behaviors is helpful for establishing a state machine based on memory operation behaviors. Then, we can analyze state transition of the state machine in the program to determine whether a memory leak defect exists. The memory operation behaviors include the following three types:

#### 3.2.1. Allocate

We use the defect pattern matching algorithm to determine whether the memory allocation exist. If the answer is yes, the state machine that belongs to this memory is established, and then the state of the state machine is analyzed to detect any memory leaks. An abstract description of the allocating memory behavior is expressed as
(1)$$\begin{array}{c} \mathrm{Alloced}\left(p\right)=\left(v_{1}\left(v\right),w_{1}\right),\left(v_{2}\left(v\right),w_{2}\right),\left(v_{3}\left(v\right),w_{3}\right),\ldots ,\left(v_{n}\left(v\right),w_{n}\right). \end{array}$$
where Alloced(p) represents a collection of memory allocation behaviors that occur within the program. $\left(v_{i}\left(v\right),w_{i}\right)$ represents a memory allocation, in which $v_{i}\left(v\right)$ represents a collection of variables that have the ownership of the memory space. If $v_{i}\left(v\right)$ is empty, the memory may have been released or a memory leak defect has occurred. $w_{i}$ indicates the function for this memory allocation.

#### 3.2.2. Free

Free refers to the process of memory release by the functions (free,delete,delete[]) provided by C/C++. An abstract description of the memory release is expressed as
(2)$$\begin{array}{c} \mathrm{Free}\left(p\right)=\left(v_{1}\left(v\right),f_{1},\;\mathrm{record}\;\right),\left(v_{2}\left(v\right),f_{2},\;\mathrm{record}\;\right),\ldots ,\left(v_{n}\left(v\right),f_{n},\;\mathrm{record}\;\right). \end{array}$$
where Free(p) represents a collection of memory release behaviors that occur within the program. $\left(v_{i}\left(v\right),f_{i},\mathrm{record}\right)$ represents the memory release. $f_{i}$ represents the function for memory release, where $f_{i}\in \left\{free,delete,delete\left[\right]\right\}$. A match check will be performed for $f_{i}$ and $w_{i}$ to check the correctness. The parameter record indicates whether the memory space has been released.

#### 3.2.3. Transfer

Transfer refers to the change in the variable that has access to the memory space. This change can be addition, reduction, and transfer. The behaviors of variable assignments, function returns, pointer calculations will cause $v_{i}\left(v\right)$ to change. Memory transfer occurs in the following situations: (1) Scope change: The scope of the local variable ends. For example, a variable is returned as a function return value. (2) Variable assignment. (3) Pointer calculation: When a pointer with the ownership of the memory space is calculated (such as “++”, “–”), the relevant variables are removed from $v_{i}\left(v\right)$. (4) Function call. Memory operations exist in the function. A specific analysis of the function summary is needed.

When $v_{i}\left(v\right)$ is empty, additional operations in the memory space cannot be performed, which indicates that the memory cannot be released, that is, a memory leak defect exists.

## 4. Our Proposed MLD Scheme

We will describe the detection method in this section. Based on the previous memory leak defect modes, we establish the corresponding state machine. Using the defect pattern matching algorithm, we can detect the memory operation behavior in the program. We control the state change of the state machine to discover the memory leak. To address the function call problem that occurs during the detection process, we propose a function summary generation algorithm for memory operation behavior.

### 4.1. Defect Pattern Matching

This matching involves matching the abstracted defect pattern with the lexical unit doubly linked list to identify suspicious defect points. We designed a fuzzy matching algorithm based on regular expressions. As shown in Algorithm 1.

The abstraction of a single lexical unit is shown in Table 1. The time complexity of Algorithm 1 is O(d*n), where *n* represents the length of the doubly linked list after the program is preprocessed; it is primarily determined by the size of the program. d represents the product of the number num of the matching string units and the longest length len of the units.
**Algorithm 1:** Match(TokenList, Pattern)**Input** **:**TokenList, Pattern**Output** **:**firstP: Record the location when the match was successful. If the match was not successful, it is null.endP: Record the next position of the end point when the match is successful, which is also the starting point of the next match.1:firstP=null,endP=null,set=null2:**while**token in TokenList **do**3:   **while** each subP in Pattern **do**4:     curPattern=getCurPattern(subP)5:     **if** curPattern is Single Selection Unit **then**6:        addSet(set,curPattern)7:      **else if**curPattern == Multiple Selections Unit **then**8:        **if** expressions(a):‘‘[abc]" **then**9:          multiPattern1(set,curPattern)10:        **else if**expressions(b)orexpressions(c)
**then**11:          multiPattern1(set,curPattern)12:        **end if**13:     **end if**14:     **if** compare(token,set) **then**15:        **if** firstP==null **then**16:          firstP=token17:        **end if**18:        token=getNextToken(TokenList)19:     **end if**20:     subP=getNextPattern(Pattern)21:     set=null22:     **if** (token==null&&subP==null)||(token!=null&&subP=null) **then**23:        endP=token24:        check(firstP,endP)25:        **else if**token==null&&subP!=null
**then**26:        contine;27:     **end if**28:     subP=getFirst(Pattern)29:     firstP=null30:   **end while**31:  **end while**32:**return**firstP,endP

### 4.2. State Machine Construction

We establish a state machine based on the memory operation behavior. We use the proposed matching algorithm to identify the memory operation behavior in the code and analyze any related defects according to the state transition of the state machine. The memory operation behavior state machine is shown in Figure 4.

### 4.3. Function Summary Generation

#### 4.3.1. Function Summary Definition

The memory operation behaviors may occur in multiple functions. For example, memory allocation is in function *a*, and the memory is freed in function *b*. Therefore, the memory operation behaviors across functions must be analyzed during the detection process, which may cause a repeated scan of functions and it is an extremely inefficient method. To solve this problem, we introduce a function summary technique to memory leak detection. The function summary describes the memory operation behaviors within the function. When a function call statement is detected, the corresponding function summary can be directly employed to obtain the memory operation behaviors to improve the detection efficiency.

The function summary for the memory operation behaviors is an abstract description of the memory operation behaviors within a function. This summary is generally expressed as a triple $<v_{i}\left(v\right),PathC,B>$, where $v_{i}\left(v\right)$ represents the set of variables that have ownership of the memory space; PathC indicates the external conditions of the behavior; and *B* represents the memory operation behaviors in this function. The abstract description of the memory operation behavior is <behavior, supplementary description of this behavior>.

#### 4.3.2. Function Summary Generation

To obtain the corresponding summary information, we generate and check a Function Call Graph (FCG) of the source code to determine whether a ring exists. The existence of the ring is essentially an incorrect call relationship, which indicates the possibility of an infinite loop in the code, and the defect may be reported. We adopt a bottom-up traversal strategy for the FCG to collect the leaf nodes in the graph and generate a CFG. We perform the data flow analysis on the nodes on the CFG, and abstract the nodes information with the memory operation behaviors to obtain the function summary. The function summary generation algorithm is designed as Algorithm 2.
**Algorithm 2**: SummaryGeneration (CFG)**Input** **:**CFG**Output** **:**summaryFunc1:**for** each curNode in CFG **do**2:   preNodes=getPreSet(curNode)3:   nr=null4:   **while** preN in preNodes **do**5:     nr=join(preN,nr)6:   **end while**7:   np=trans(nr,curNode)8:   **if** memoryoperation on np **then**9:     oper=getMemoryOper(np)10:     update(summaryFunc)11:   **end if**12:**end for**13:**return**summaryFunc

The time complexity of Algorithm 2 is O(n∗count), where *n* denotes the number of nodes on the CFG, and count denotes the maximum number of precursor nodes.

#### 4.3.3. Function Summary Update

Algorithm 2 pertains to the case without a function call. To obtain summary information about all functions in the program to be detected, we should obtain the summary information of the bottom nodes on the FCG, and then perform a bottom-up update of the summary layer by layer are necessary. The error is handled when the FCG has rings. The function summary update algorithm (Algorithm 3) addresses the case in which no rings exist on the FCG.
**Algorithm 3**: SummaryUpdate (FCG)**Input** **:**FCG**Output** **:**allSummaryFunc1:genList=getLeafNode(FCG)2:**while**!(genList.isempty())**do**3:   genFunc=getFunc(genList)4:   **if** isUpdateSubFunc(genFunc) **then**5:     summaryFunc=getSummaryFunc(genFunc)6:     allSummaryFunc.add(summaryFunc)7:   **else**8:     genList.add(genFunc)9:   **end if**10:   **for** function *f* calls genFunc on FCG **do**11:     **if** *f* is not in genList **then**12:        genList.add(f)13:     **end if**14:   **end for**15:**end while**16:**return**summaryFunc

The time complexity of Algorithm 3 is divided into two parts: a summary generation algorithm for a single function O(n∗count) and the number countFunc of functions on the FCG, where *n* represents the maximum number of precursor nodes of the CFG. Thus, the time complexity of Algorithm 3 is O(n∗count∗countFunc).

### 4.4. Memory Leak Detection

#### 4.4.1. General Memory Leak Detection

General memory leak detection consists of memory leak detection in a function with no call relationships or multiple call relationships. The core idea is establishing a state machine based on memory operation behaviors and control of the changes of the state machine. The state machine is established by obtaining the memory allocation statement and is directly changed from the initial “Start” to “Alloced”. We combined the fuzzy matching algorithm with the function summary information to analyze the subsequent statements to control the state machine. For a function statement analysis, we handle statements in functions that may have memory operations, such as assignments, memory allocation, memory free, returns, function calls, and if-else branches. The general memory leak detection algorithm is described as Algorithm 4.
**Algorithm 4**: GeneralCheck (program)**Input** **:**program**Output** **:**memoryLeakReport1:genSummary(program)2:**for** each func in program **do**3:   **if** Visit(func) **then**4:     continue5:   **end if**6:   nodeCFG=genCFG(func)7:   BFS(nodeCFG)8:**end for**9:**return**memoryLeakReportBFS(nodeCFG)**Input** **:**nodeCFG**Output** **:**memoryLeakReport2:**while** each node in CFG is not null **do**   **if** node is operation about memory **then**4:     update(StateMachine)     **if** StateMachine is ‘‘ERROR" **then**6:        memoryLeakReport.add()     **end if**8:     **else if**
node is function call **then**     getSummary(func)10:     update(StateMachine)     **if** StateMachine is ‘‘ERROR" **then**12:        memoryLeakReport.add()     **end if**14:    **else if**
node is conditional branch **then**     **for** each nodeBranch in conditionalbranch **do**16:        BFS(nodeBranch)     **end for**18:   **end if****end while**20:**return**memoryLeakReport

The time complexity of Algorithm 4 is $O(n^{3}*countFunc)$, where countFunc represents the number of functions on the FCG and *n* represents the maximum number of precursor nodes of the CFG.

#### 4.4.2. Special Memory Leak Detection

Special memory leak detection is a special rule detection for C++ [22], which is the memory leak detection of class members summarized in the memory leak defect modes.

If the memory leak is caused by the inconsistency of the constructor and the destructor; first, we need to obtain all class definitions. For each class, we check the constructor for the memory allocation behaviors of the class members. If it exists, we generate an abstract description of the corresponding memory allocation behavior. Second, we check the destructor for the memory release behaviors of the class members; if it exists, we generate an abstract description of the corresponding memory release behavior. Last, we match the abstract description of the memory release behavior against the abstract description of the memory allocation behavior, and the reporte mismatch.

If the destructor of the parent class with the subclass is not virtual. First, we need to obtain the parent-child relationship of all classes in the detection program. Second, we check the destructor of each parent class, if it does not exist or it is not virtual, we reporte an error.

If the copy constructor or assignment overloaded function is not properly defined. First, we need to obtain all class definitions in the test program. Second, we need to determine whether a class member variable points to the memory space in the class and whether the class has a custom copy constructor or assignment overload function. Last, we have to check whether an improper object copy modes (shallow copy) is used to construct the object.

A special memory leak detection algorithm is described as Algorithm 5.The time complexity of Algorithm 5 is O(num∗n), where num represents the number of classes defined in the program, and *n* represents the detection time for a single class.
**Algorithm 5**: SpecialCheck (program)**Input** **:**program**Output** **:**memoryLeakReport1:classInfo=getClassInformation(program)2:relationMap=getRelation(classInfo)3:**for**parentClass in relationMap **do**4:   **if** !checkVirtual(parentClass) **then**5:     memoryLeakReport.add()6:   **end if**7:**end for**8:**for** each class in classInfo **do**9:   allocInfo=getAlloc(class)10:   freeInfo=getFree(class)11:   **if** !match(allocInfo,freeInfo) **then**12:     memoryLeakReport.add()13:   **end if**14:   cpInfo=checkCopyorAssignment(class)15:   **if** (cpInfo is error) || (cpInfo==NULL && default copy constructor or assignment function) **then**16:     memoryLeakReport.add()17:   **end if**18:**end for**19:**return**summaryFunc

## 5. Performance Analysis

Based on the designed memory leak detection method, we implement a static detection system named ZkCheck. The experimental environment of this paper is shown in Table 2.

The dataset we use contains codes of some mainstream frameworks or algorithms written in C/C++ language, all of which are open source software. (all website accessed on 4 July 2022 ) Gcc (http://www.gnu.org/software/bash/) is written as the compiler for the GNU operating system. Ammp (https://github.com/rob-n3ikc/ammp) is a simulation software for molecular dynamics research. Bash (https://gcc.gnu.org/gcc-12/) is the GNU Project’s shell—the Bourne Again SHell. It is written as a command processor. Mesa (https://www.mesa3d.org/) is an open source software implementation of OpenGL, Vulkan, and other graphics API specifications. Cluster (https://github.com/pthimon/clustering) is a tiny c++ project which can automatically fidn the ideal number of clusters of an affinity matrix. OpenCV (Open Source Computer Vision Library) (https://opencv.org/releases/) is an open source computer vision and machine learning software library. Bitcoin (https://github.com/bitcoin/bitcoin) is a software written in C++ language that fully validate blocks and transactions.

### 5.1. Evaluation Indicators

To evaluate the detection efficiency of static detection tools, we employ the False Positive Rate, False Negative Rate and detection time as evaluation indicators. We represent the total number of defects detected by static detection as *C*; the program should have the total number of defects, which is denoted as $act_{-}C$; and the number of false positives is FC. The definitions of False Positive Rate (FPR) and False Negative Rate (FNR) are expressed as follows:(3)$$\mathrm{FNR}=\frac{\left|act_{-}C-C\right|}{act_{-}C}.$$

The calculation formula of FPR is expressed as follows:(4)$$\mathrm{FPR}=\frac{FC}{C}.$$

Since obtaining the actual number of defects in the source program to be detected $act_{-}C$ is difficult, the False Positive Rate is the main focus of the experimental results in this paper. The False Negative is overlooked in this experiment.

### 5.2. Experiments and Result Analysis

#### 5.2.1. Validation Experiment

The experimental steps are described as follows. First, we obtain seven open source mainstream software and their codes. We perform statistics on size and bugs. Then, we take the code of each software as input and analyze the results using our proposed algorithm ZkCheck. Finally, we evaluate our algorithm based on the experimental results.

Due to the lack of multimedia software, and we cannot get the actual number of defects in those softwares (gcc, ammp, bash, mesa, cluster, openCV and bitcoin). We use the proposed algorithm to detect the open source projects in Table 2, and test the defects to obtain the number of False Positives. We use manual verification methods to ensure the accuracy of the above data. The people checking the code are graduate students in school of computer who are familiar with C/C++ language. The experimental results are shown in Table 3.

The experimental results show that the detection speed is approximately 1.1 Kloc/s, and the False Positive Rate is approximately 23.8%. By the analysis of the experimental results, we further verify the feasibility of the static memory leak detection method based on the defect modes.

As shown in Figure 5 and Figure 6, we observe the relation between the code quantity and the detection time is approximately a straight line, which means a positive correlation between the number of code lines in the open source project and the required detection time. The larger is the code of open source project, the longer is the time required to detect the code.

Figure 7 indicates that there is no correlation between the code quantity and the number of reported defects or the number of false positives. As the amount of code increases, the number of reported defects may increase or decrease, which is primarily determined by the project source code. Comparing the open source project ammp with mesa, ammp has substantially less code but a larger number of reported errors than mesa.

#### 5.2.2. Contrast Experiment

The compared detection tools, the detection time and the number of reported defects and false reports are shown in Table 4.

Figure 8 and Figure 9 show the comparison results of the detection tools for detection time and False Negative Rate, respectively. An analysis of the experimental results reveals that our method has the fast and stable detection speed, which is superior to Clouseau, CSA, Sparrow and other tools. In terms of the False Negatives Rate, our method is lower than Clouseau, similar to CSA, and higher than Sparrow. In terms of the number of reported defects, our method is superior to CSA and Sparrow.

The result shows that the function summary to reduce the repeated spread detection problems in the function call points improves the detection speed and reduces the detection time. The results of Experiment 2 show that ZkCheck has a relatively satisfactory comprehensive efficiency compared with other detection tools and improves its detection speed based on ensuring a certain low False Positive Rate.

#### 5.2.3. Comprehensive Analysis

The above experiment show that ZkCheck has a relatively satisfactory comprehensive efficiency compared with other detection tools and improves its detection speed based on ensuring a certain low False Positive Rate. The dateset of codes are mainstream software of image processing, IoT and industrial modeling. Therefore, the memory leak detection algorithm we proposed is cpmpatible and can be widely used in the detection of major actual software and other industrial software fields.

## 6. Industrial Applications

With the increasing popularity of smart devices and 5G networks in our daily life, multimedia data acquisition, storage and processing on the Internet have become increasingly convenient. In the operation of large-scale real-time software such as multimedia software, memory leaks are very common in memory defects, usually caused by improper management of dynamic memory in program implementation [23]. When the dynamically allocated memory in the program is not released in time, there will be leakage, which will cause the memory space to be consumed and cannot be recycled and reused. it can result in a reduction in memory leakage resources, and ultimately reduce program performance, resulting in the overall software system collapse. At the same time, memory leak has a certain degree of concealment, it is easy to be ignored. Especially for the language of C/C++, the expansion of the expansion scale and the programmer’s irregular programming are more common.

Our detection algorithm ZkCheck is a static detection algorithm, which can detect errors in the software before it runs. At the same time, experiments show that our algorithm performs better than other detection algorithms in terms of detection speed and accuracy. This advantage shows that in the industrial field with strict standards, ZkCheck can predict abnormality before the machine starts, therefore reducing risks and ensuring engineering safety.

## 7. Conclusions and Future Work

In this paper, we present a static method for memory leak detection based on a defect modes. This method statically detects leaks in C/C++ programs. We preprocess the source code, perform lexical analysis, and convert it into a two-way linked list with language keywords as the unit. We add corresponding symbol table information to the two-way linked list. By matching the two-way linked list with the corresponding defect modes to control the change in the defect state machine, we realize memory leak defect detection. By comparing the existing detection tools, our experimental results show that the static detection method of memory leak based on the defect modes has a good perform in discovering potential memory leak defects in program source code, which can effectively improve the correctness of developed software. The memory leak detection algorithm we proposed can be widely used in the detection of major actual software. The method also has some limitations. When the code quantity of software is small, the rate of false positives might increase drastically. When the software scale reaches a higher level, a further increase in speed is required.

For the future work, we will explore the following aspects:(1)Provide corresponding solutions for the defects detected by the algorithm.(2)Path-sensitive processing. Path explosions may occur due to excessive conditional statements, loop statements and function recursions in the encoding. Therefore, effective analysis of the convergence of the path nodes based on ensuring efficiency is the focus of future research.(3)Improve the accuracy of detection. For static detection, the inaccuracy of interval arithmetic in the detection process may cause a large number of False Negatives or False Positives.

## Figures and Tables

**Figure 1 entropy-24-00947-f001:**
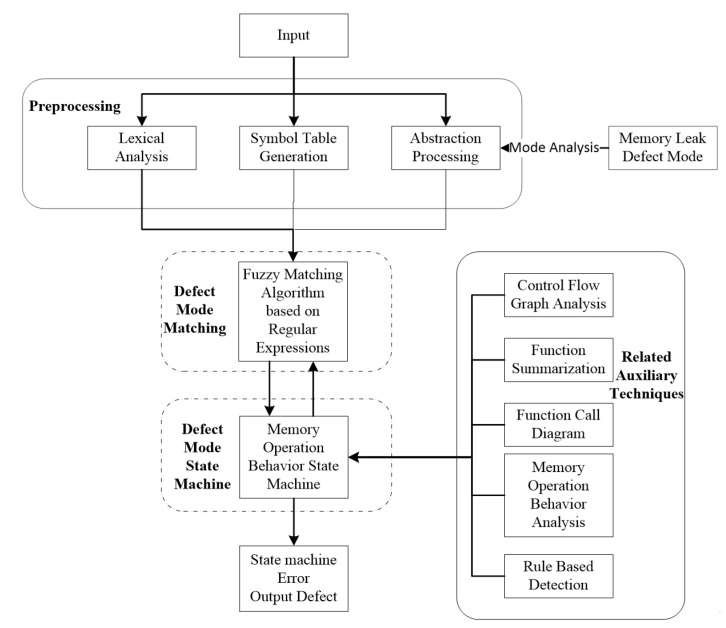
Static detection architecture diagram based on defect mode.

**Figure 2 entropy-24-00947-f002:**
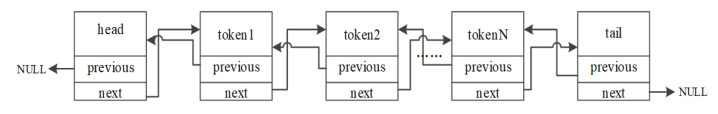
Doubly linked list for lexical units.

**Figure 3 entropy-24-00947-f003:**
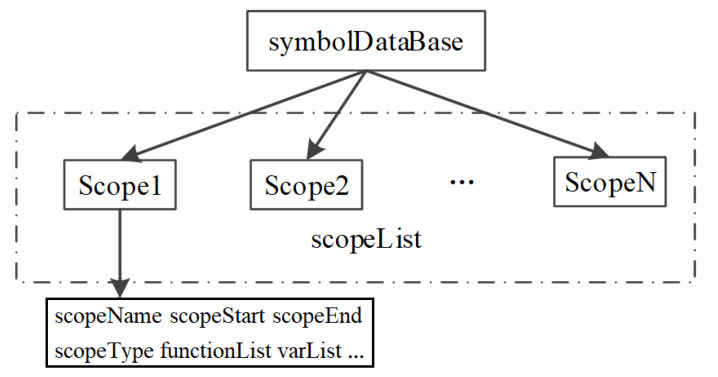
Structure of the symbol table.

**Figure 4 entropy-24-00947-f004:**
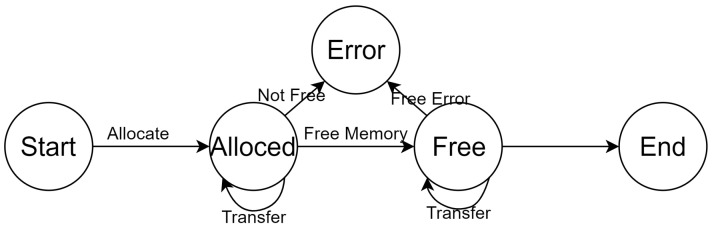
Memory operation behavior state machine.

**Figure 5 entropy-24-00947-f005:**
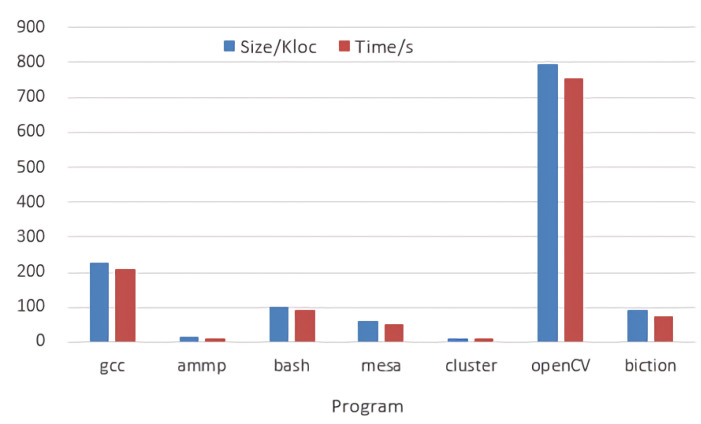
Comparison of code quantity and detection time.

**Figure 6 entropy-24-00947-f006:**
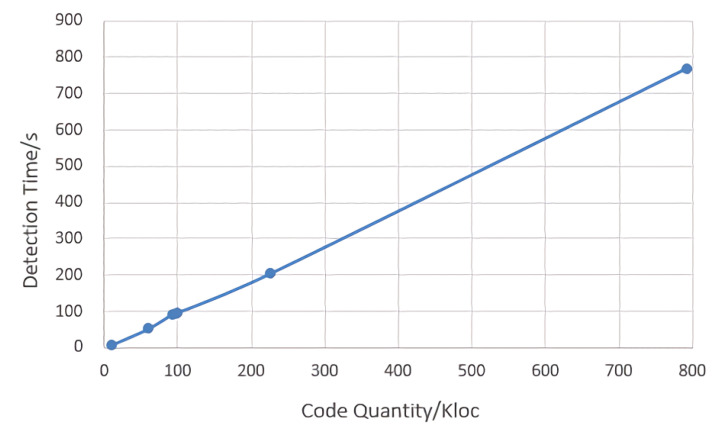
Relation between code quantity and detection time.

**Figure 7 entropy-24-00947-f007:**
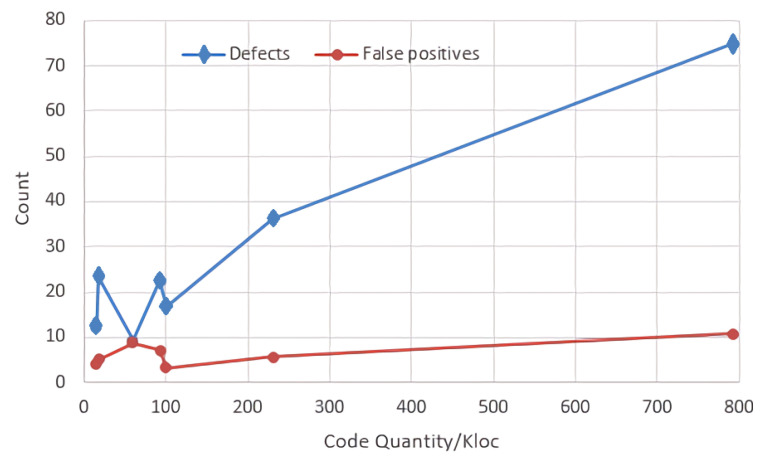
Comparison of code quantity with the number of reported defects and the number of false positives.

**Figure 8 entropy-24-00947-f008:**
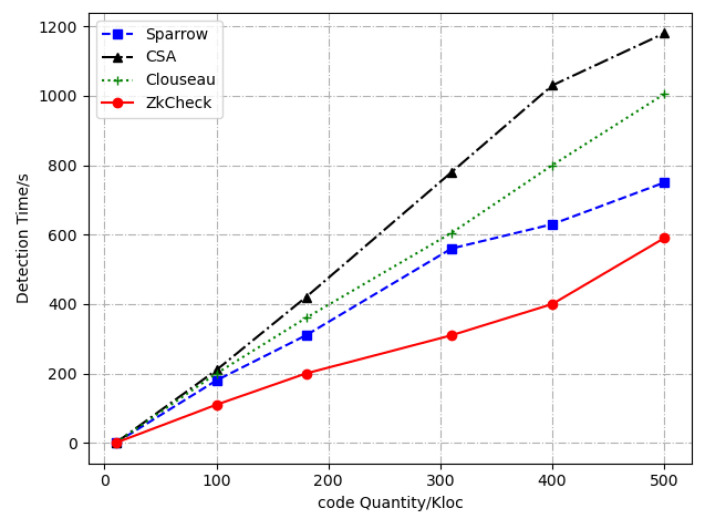
Comparison of detection tools for detection time.

**Figure 9 entropy-24-00947-f009:**
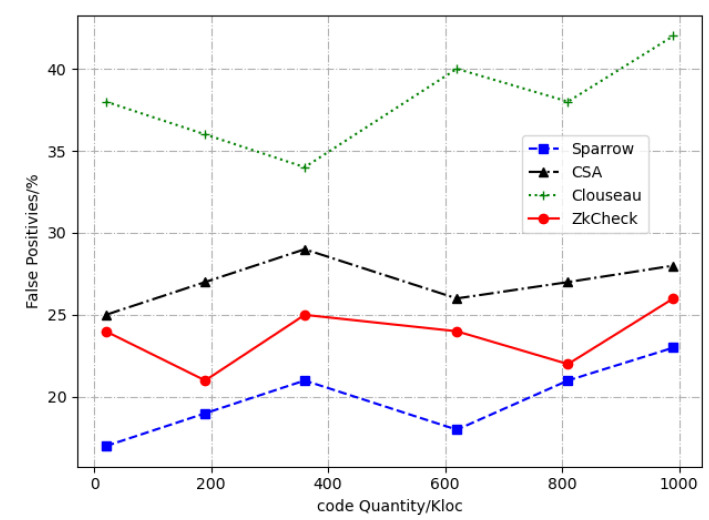
Comparison of detection tools for false positive rate.

**Table 1 entropy-24-00947-t001:** Explanation of abstraction of defect pattern.

Symbol	Meaning
%any%	Any form of keywords, which can be any lexical units, such as variables, types, and operators.
%name%	A variable name or type. For example, “int a” can be expressed as “%name% %name%”.
%type%	A variable type. For example, “int a” can be expressed as “%type% %name%”.
%num%	A number, such as “23”.
%bool%	A Boolean value, “true” or “false”
%comp%	A comparison operator, such as “>”, “==”, etc.
%str%	A string.
%var%	A variable.
%varid%	A variable ID.
%or%	“|”
%oror%	“||”
%op%	An operator, such as “=”.

**Table 2 entropy-24-00947-t002:** Experimental enviroment.

OS	ubuntu14.04, windows
memory	4G
cpu	X64 (i5)
Comparison Method (memory leak detection tool)	Clang Static Analyzer, Sparrow, Clouseau
Dataset (open source C\C++ project)	gcc, ammp, bash, mesa, cluster, openCV, bitcoin

**Table 3 entropy-24-00947-t003:** Results of validation experiment.

Program	Size (Kloc)	Times (s)	Bug Count	False Count
gcc	230.4	213.1	36	6
ammp	13.4	10.4	23	5
bash	100.0	90.1	16	3
mesa	61.3	48.6	9	8
cluster	10.7	9.5	12	4
openCV	794.6	756.8	74	11
bitcoin	94.4	78.7	22	7
Total	1304.8	1257.9	192	44

**Table 4 entropy-24-00947-t004:** Results of experiment 2.

Leak Detector	Speed (loc/s)	Bug Count	False Positive Rate (%)
Clang Static Analyzer (CSA)	400	81	27
Sparrow	720	69	19
Clouseau	500	92	41
ZkCheck	1137	102	24

## Data Availability

Not applicable.

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
