# Peer review of "MLD: An Intelligent Memory Leak Detection Scheme Based on Defect Modes in Software"

_entropy, 2022, doi:10.3390/e24070947_

Round 1
Reviewer 1 Report
The work describes a method to identify potential sources of memory leaks in C/C++ code. The formulation of the paper is nice. A major weakness is the experimental evaluation, as the establishment of the ground truth and the exact way the proposed approach are compared to others is not clear.
More specific comments:
It is not clear why so much emphasis (in the title, abstract, introduction and conclusions) is given to multimedia. The core of the paper seems to be generic and applicable to any C/C++ code without any specific features targeting multimedia software. Moreover, none of the presented experiments include multimedia software.
Section 1, "However, this method also has the disadvantages of slow detection speed and 30
serious false positives." --> A bibliographic reference is needed to support this statement, as it is central to establishing the motivation for the presented work.
Section 2, "There disadvantages" --> Their disadvantages
Section 5. The establishment of the ground truth is not explained and is needed. For example, in the first line of Table 3, how do the authors differentiate between the 36 true bugs and the 6 false positives? And where are the false negatives? These are also needed, since they are discussed in the motivation of the work.
Section 5. For 5.2.1 it is explained that no multimedia software is available and other projects are used, listed in table 3 and results are presented separately for each project as well as cumulatively. Then, for 5.2.2 where other detectors are also evaluated, there is no mention of what is used as input test. The results given from the authors' approach do not match neither the individual nor the total results from 5.2.1. So, how exactly is the experiment set up?
Section 5. 5.3 does not contain any performance analysis but rather a discussion. It does not belong in section 5. It can be either part of the conclusions or a separate section.
Author Response
Point1: It is not clear why so much emphasis (in the title, abstract, introduction and conclusions) is given to multimedia. The core of the paper seems to be generic and applicable to any C/C++ code without any specific features targeting multimedia software. Moreover, none of the presented experiments include multimedia software.
Response1: This paper is related to a previous project on data transfer on smart devices. The transmitted data includes structured and unstructured multimedia information. In the process of developing multimedia software, we study how to analyze software defects automatically.
Point2: Section 1, "However, this method also has the disadvantages of slow detection speed and 30
serious false positives." --> A bibliographic reference is needed to support this statement, as it is central to establishing the motivation for the presented work.
Response2: We corrected this mistake by using references in recent 5 years.
Point3: Section 2, "There disadvantages" --> Their disadvantages
Response3: We corrected this spelling mistake.
Point4: Section 5. The establishment of the ground truth is not explained and is needed. For example, in the first line of Table 3, how do the authors differentiate between the 36 true bugs and the 6 false positives? And where are the false negatives? These are also needed, since they are discussed in the motivation of the work.
Response4: We added details of dataset according to your reviews. The true bugs are detected by compiler. Our algorithm detects "positives". Some of "positives" are true bugs, while others are classified as false positives. The people checking the code are graduate students in school of computer who are familiar with C/C++ language.
Point5: Section 5. For 5.2.1 it is explained that no multimedia software is available and other projects are used, listed in table 3 and results are presented separately for each project as well as cumulatively. Then, for 5.2.2 where other detectors are also evaluated, there is no mention of what is used as input test. The results given from the authors' approach do not match neither the individual nor the total results from 5.2.1. So, how exactly is the experiment set up?
Response5: We added details of experiment.
The experimental steps are described as follows. First, we obtain seven open source mainstream multimedia software and their codes. We perform statistics on size and bugs. Then, we take the code of each software as input and analyze the results using our proposed algorithm ZkCheck. Finally, we evaluate our algorithm based on the experimental results.
Point6: Section 5. 5.3 does not contain any performance analysis but rather a discussion. It does not belong in section 5. It can be either part of the conclusions or a separate section.
Response6: Industrial application belongs to section 5 according to the template. It explains the application effect and some prospects of our algorithm in the industrial domain.
Reviewer 2 Report
This article proposes a Memory Leak Detection Scheme based on Defect Modes in Multimedia Software. The authors propose a solid proof of concept.
The authors should address the minor issues I have noted in their submitted article (Please refer to my comments on the document.)
The authors should thoroughly revise their article for grammar and sentence building. Please fix the highlighted words and sentences on your submitted manuscript.

Author Response
Point 1: The authors should address the minor issues I have noted in their submitted article (Please refer to my comments on the document.) The authors should thoroughly revise their article for grammar and sentence building. Please fix the highlighted words and sentences on your submitted manuscript.
Response 1: Thank you for providing these valuable comments on our paper, we will revise the paper as soon as possible and send you the revised version. .
Reviewer 3 Report
In this manuscript, the authors propose a framework for intelligent memory leak detection in multimedia software. They combine fuzzy matching and function summary methods to determine the behavior of memory operations and assess the source code vulnerability. The obtained experimental results show the effectiveness of the proposed methodology for memory leak detection and its quality (detection speed and accuracy).
My remark is as follows:
In the abstract and “1. Introduction” section, the novelty and authors’ contributions in the quality assurance of multimedia software should be emphasized.
Please add some details about the datasets, mentioned in “5. Performance Analysis” section.
In “5.2. Experiments and Result Analysis” subsection, there are some repetitions. For example, Figure 5 – Figure 7 repeat data from Table 3, Figure 8 – Figure 9 – from Table 4 respectively.
Please enhance this subsection adding a comprehensive comparison with results obtained using similar existing methods.
What are the limitations of your study?
In ‘References’, please, include references about similar previous studies from the last five years.
Technical remarks:
Please unify the abbreviations of measurement units: “sec” or “s”, “KLOC” or “Kloc”.
Link to the datasets could be added.
Author Response
Point1: In the abstract and “1. Introduction” section, the novelty and authors’ contributions in the quality assurance of multimedia software should be emphasized.
Response1: I think the original paper showed the novelty and contributions. "We propose an intelligent memory leak detection scheme based on defect modes MLD in multimedia software." We summarized memory operation behaviors and divided them into three types based on the memory leak defect modes.
Point2: Please add some details about the datasets, mentioned in “5. Performance Analysis” section.
Response2: We added description and links of each dataset in the revised paper.
Point3: In “5.2. Experiments and Result Analysis” subsection, there are some repetitions. For example, Figure 5 – Figure 7 repeat data from Table 3, Figure 8 – Figure 9 – from Table 4 respectively.
Please enhance this subsection adding a comprehensive comparison with results obtained using similar existing methods.
Response3: We adjusted the structure of the paper and added a subsection of comprehensive analysis.
Point4: What are the limitations of your study?
Response4: We added limitations in the conclusion section. The method also has some limitations. When the code quantity of multimedia is small, the rate of false positives might increase drastically.
Point5: In ‘References’, please, include references about similar previous studies from the last five years.
Response5: We added references about similar studies in the last five years and referred to them in "related work" section.
Point6: Technical remarks:
Please unify the abbreviations of measurement units: “sec” or “s”, “KLOC” or “Kloc”.
Response6: Thank you for your comments. We corrected all related errors in the paper.
Point7: Link to the datasets could be added.
Response7: We added description and links of each dataset in the revised paper.
Round 2
Reviewer 1 Report
This is an ok paper on detecting memory leaks in C/C++ code.
The authors have addressed all other comments but one, and have done so more than adequately. The one concern that remains is the one on whether the work is indeed related to multimedia software.
In their response the authors explain that the background of their work comes from a project where they worked with multimedia software. That alone does not make the current work related to multimedia software as well. In fact, there is nothing in the paper to connect it to multimedia. This is particularly true in the experimental results, as all the presented applications come from other domains.
In their response the authors state "First, we obtain seven open source mainstream multimedia software and their codes. " I am very sorry to see this statement; as academics we may sometimes see things differently, but we at least support our position with rational arguments and always with the truth. Among the 7 software used in the experiments, 5 (GCC, Ammp, Bash, Cluster, Bitcoin) have no relation whatsoever to multimedia.
If the authors insist in presenting their work as related to multimedia, then only experiments on Mesa and OpenCV are related, the others need to be removed. If, on the other hand, the authors insist in preserving all experiments (personally I would suggest this option), then the title/abstract/introduction need to be updated to show the more generic nature of the work.
Author Response
Point1: The authors have addressed all other comments but one, and have done so more than adequately. The one concern that remains is the one on whether the work is indeed related to multimedia software.
Response1: Thank you for your detailed comments. The reason why we have always moved our research and experiments to the field of multimedia software is to get close to the theme of journal submissions.
I want to know if we remove the multimedia words in the title and content and revise carefully, will the paper be possible to be accepted in the end?
Round 3
Reviewer 1 Report
I have no further comments on this work